# Real-World Incidence of Immune-Related Adverse Events Associated with Nivolumab Plus Ipilimumab in Patients with Advanced Renal Cell Carcinoma: A Retrospective Observational Study

**DOI:** 10.3390/jcm10204767

**Published:** 2021-10-18

**Authors:** Satoshi Washino, Hideki Takeshita, Masaharu Inoue, Makoto Kagawa, Takahiko Soma, Hodaka Yamada, Yukio Kageyama, Tomoaki Miyagawa, Satoru Kawakami

**Affiliations:** 1Department of Urology, Jichi Medical University Saitama Medical Center, Saitama 330-0834, Japan; sh2-miya@jichi.ac.jp; 2Department of Urology, Saitama Medical Center Saitama Medical University, Kawagoe 350-8550, Japan; take_uro@yahoo.co.jp (H.T.); mkt61307ms@gmail.com (M.K.); kawakami@saitama-med.ac.jp (S.K.); 3Department of Urology, Saitama Cancer Center, Saitama 362-0806, Japan; m-inoue@saitama-pho.jp (M.I.); t.soma.uro@gmail.com (T.S.); kageyamauro@cancer-c.pref.saitama.jp (Y.K.); 4Department of Endocrinology, Jichi Medical University Saitama Medical Center, Saitama 330-0834, Japan; hyamada0510@hotmail.co.jp

**Keywords:** nivolumab, ipilimumab, hypopituitarism, thyroid toxicities, adrenal insufficiency, renal cell carcinoma, immune-related adverse events

## Abstract

Real-world incidence of immune-related adverse events (irAEs) associated with nivolumab plus ipilimumab in patients with renal cell carcinoma (RCC) has been rarely demonstrated. The present study aims to report the safety outcomes of this combination therapy in the real-life population. We conducted a multi-institutional retrospective observational study that assessed the incidence and severity of irAEs associated with nivolumab plus ipilimumab in 41 Japanese patients with metastatic and/or locally advanced RCC. The irAEs were classified into endocrine and non-endocrine irAEs. The median age and follow-up period were 68 years and 13.0 months, respectively. Endocrine irAEs were observed in 66% of patients, including hypopituitarism in 44%, hyperthyroidism in 41%, and primary hypothyroidism in 22%, while non-endocrine irAEs were observed in 54%. All patients experiencing hypopituitarism presented with adrenocorticotropic hormone deficiency, causing secondary adrenal insufficiency, which required permanent corticosteroid replacement therapy. There was an association between the incidence of endocrine irAEs and high-grade non-endocrine irAEs other than skin-related irAEs (*p* = 0.027). When patients experienced two or more endocrine irAEs, they had a 35% chance of experiencing high-grade non-endocrine irAEs other than skin-related irAEs. Nivolumab plus ipilimumab may lead to a high prevalence of endocrine irAEs in “real-world” patients. Endocrine irAEs may be associated with non-endocrine irAEs other than skin-related irAEs.

## 1. Introduction

In recent years, immune checkpoint inhibitors (ICIs) have become a mainstay treatment for a wide variety of cancers, including renal cell carcinoma (RCC). Antibodies that target programmed cell death protein 1 (PD-1) or one of its ligands, programmed death-ligand 1 (PD-L1) (e.g., nivolumab, pembrolizumab, atezolizumab, avelumab, durvalumab), are active against a broad range of malignancies and are changing the treatment landscape for cancer. Ipilimumab is a monoclonal antibody directed against cytotoxic T-lymphocyte antigen-4 (CTLA-4) [1]. CheckMate 214, a phase 3 clinical trial comparing nivolumab plus ipilimumab with sunitinib for previously untreated clear cell advanced RCC, demonstrated that overall survival and objective response rates were significantly higher with nivolumab plus ipilimumab than with sunitinib, among intermediate- and low-risk patients [2]. Nivolumab plus ipilimumab is approved as a first-line treatment for RCC [2] and is also used for the treatment of melanoma and non-small cell lung cancer [3,4]. Due to their targets and mechanisms of action, ICIs can cause autoimmune and inflammatory effects, termed immune-related adverse events (irAEs), which most commonly affect the skin, gastrointestinal tract, liver, and endocrine system [5,6]. The pituitary, thyroid, and adrenal glands are the endocrine organs typically affected by immune checkpoint blockade [7]. Although irAEs associated with PD-1/PD-L1 blockade are similar to those associated with CTLA4 blockade (ipilimumab), the incidence and severity of irAEs differ between PD-1/PD-L1 blockade and CTLA4 blockade and are higher with combination blockade of PD-1/PD-L1 and CTLA4 compared to blockade of each separately [7,8].

The CheckMate 214 trial demonstrated that treatment-related AEs of any grade and Grade 3–4 occurred in 93% and 46% of patients treated with nivolumab plus ipilimumab, respectively. High-dose glucocorticoids were required in 35% of patients due to irAEs [2]. In melanoma, high grade irAEs were reported in 60% of the patients treated with the combination of nivolumab plus ipilimumab, almost all of whom were exposed to steroid treatments in the real-world population [9]. However, there is limited data on safety of nivolumab plus ipilimumab therapy in RCC in the nonclinical trial setting, and only one study has assessed the irAEs associated with nivolumab plus ipilimumab in RCC in the “real world” [10].

We conducted a multi-institutional observational study to assess the incidence and severity of irAEs in metastatic or advanced unresectable RCC patients treated with nivolumab plus ipilimumab and demonstrated that the incidence of irAEs in the endocrine system was much higher than in the CheckMate 214 study.

## 2. Materials and Methods

### 2.1. Study Design and Data Collection

This multi-institutional retrospective observational study was approved by the institutional review board of Jichi Medical University Saitama Medical Center (JMU) (Saitama, Japan), Saitama Medical Center Saitama Medical University (SMU) (Saitama, Japan), and Saitama Cancer Center (SCC) (Saitama, Japan) (No. RinS20-058, 2442, and 1130, respectively). Data in patients who received nivolumab plus ipilimumab for previously untreated metastatic and/or locally advanced unresectable RCC at JMU, SMU, or SCC from September 2018 to July 2020 were collected from medical records in each institute. A total of 41 patients (19, 14, and 8 patients from SMU, SCC, and JMU, respectively) were included, and all of them were assessed in the present study.

### 2.2. irAEs

The AEs were graded according to the National Cancer Institute Common Terminology Criteria for Adverse Events, version 5.0. The irAEs were defined as inflammatory side effects that promote immune system activity and cause a unique array of inflammatory events. Endocrine irAEs included adrenal insufficiency, hypopituitarism, hypophysitis, hyperthyroidism, hypothyroidism, and diabetes mellitus (hyperglycemia), while non-endocrine irAEs included pneumonitis, colitis, hepatitis (alanine aminotransferase increased and hepatic failure), nephrotoxicities (acute kidney injury, creatinine increased, and proteinuria), arthritis, myocarditis, myositis, radiculitis, and skin-related irAEs (rash, pruritus, hypopigmentation). The irAEs and these grades being assessed in the present study are listed in the Appendix A.

### 2.3. Laboratory Testing

Biochemical tests, including thyroid-stimulating hormone (TSH) and free thyroxine (fT4) with or without free triiodothyronine (fT3), were routinely measured before the initiation of treatment. These tests were performed before every cycle or every other cycle. If clinical signs and/or symptoms consistent with endocrinopathies developed, tests of basal adrenocorticotropic hormone (ACTH), cortisol, luteinizing hormone (LH), follicle-stimulating hormone (FSH), estradiol, testosterone, and/or prolactin, in addition to TSH, fT4, and fT3, were performed depending on the signs and symptoms. In addition to blood test results, pituitary magnetic resonance imaging (MRI) imaging was performed in patients presenting with relevant clinical symptoms and abnormal hormone test results. Hypopituitarism was defined as low levels of the effector hormones (cortisol, fT4, sex hormone) relative to normal laboratory ranges, along with low or inappropriately normal pituitary hormone levels (ACTH and TSH) or low gonadotrophins (LH and FSH) for age in postmenopausal women. Hormone tests were performed as needed: the corticotropin-releasing hormone (CRH) stimulation test, thyrotropin-releasing hormone, and LH-releasing stimulation tests were performed if ACTH, TSH, and gonadotroph axis deficiencies were suspected, respectively. Hypophysitis was diagnosed based on biochemical hypopituitarism [11], clinical features, and/or radiologic findings. Hyperthyroidism was defined as low levels of TSH with normal or high levels of fT4 and/or fT3, whereas primary hypothyroidism was defined as elevated levels of TSH with normal or low levels of fT4 and/or fT3. The diagnosis of endocrine irAE was diagnosed based on the management guidelines of irAEs in endocrine organs induced by ICIs established by the Japanese Endocrine Society but also the endocrinological diagnostic test [12]. The presence and severity of endocrine irAEs were reviewed by an experienced endocrinologist (H.Y.) based on the laboratory data in each patient.

### 2.4. Objectives of This Study

The primary objectives of this study were to assess the incidence rates of endocrine and non-endocrine irAEs following nivolumab plus ipilimumab therapy. The secondary objectives were to study whether there is an association between endocrine irAEs and non-endocrine irAEs.

### 2.5. Statistical Analyses

All data are expressed as the median (range) or the mean ± SD, unless otherwise indicated. The chi-square test was used to assess the association between endocrine irAEs and non-endocrine irAEs. The Student’s t-test, Fisher’s exact test, or chi-square test was used to assess the predisposing factors for high grade irAEs. Statistical analyses were performed using GraphPad Prism software (ver. 7.0; GraphPad, La Jolla, CA, USA). *p* < 0.05 was considered statistically significant.

## 3. Results

### 3.1. Patient Characteristics and Tumor Location

The study population included 28 males and 13 females, with a median age of 68 years (44–87 years) (Table 1). Radical or partial nephrectomy prior to nivolumab plus ipilimumab was performed in 17 patients. There were 15 and 26 intermediate- and poor-risk patients, respectively, according to the International Metastatic RCC Database Consortium risk classification. Regarding the type of RCC, 32, 7, and 2 patients had clear cell carcinoma, non-clear cell carcinoma, and unknown histology, respectively. Thirty-eight patients exhibited metastatic diseases, while the other three patients did not, at the initiation of nivolumab plus ipilimumab therapy, in which two patients had extensive lymph node involvements, and one had a locally advanced unresectable renal tumor. Tumors were located in the kidneys and lungs in 25 patients each, followed by regional lymph nodes in 12, bone in 11, liver in 8, brain in 2, and other locations in 7 patients.

### 3.2. Nivolumab Plus Ipilimumab Therapy

The median follow-up duration was 13.0 months (0.1–26.5 months), and the median duration of nivolumab plus ipilimumab therapy was 2.9 months (0.1–22.1 months). A total of 27 patients (66%) received all four doses of nivolumab plus ipilimumab. Nine patients died during follow-up, whereas the other thirty-two patients were alive at the last follow-up; nine of those patients received nivolumab plus ipilimumab therapy, while fifteen were treatment-free, and eight received subsequent therapy for RCC. A total of 17 (41%) and 14 (34%) patients discontinued nivolumab plus ipilimumab temporarily or permanently due to irAEs and disease progression, respectively, while three patients discontinued these after achieving a complete or long-term partial response.

### 3.3. Endocrine irAEs

Endocrine irAEs, including hypopituitarism, hyperthyroidism, and primary hypothyroidism, were observed in 27 patients (66%). Hypopituitarism occurred in 18 patients (44%), all of whom exhibited ACTH deficiency causing secondary adrenal deficiency, and 14 of whom had Grade 3 AEs (Table 2). Serum low cortisol levels with low levels or inappropriately normal levels of ACTH were observed in 17 of 18 patients (Appendix A), while 1 patient (case SCC2 in Appendix A) exhibited relatively low levels of ACTH and cortisol at the time of onset and was later diagnosed with ACTH deficiency by a hormone test. A CRH stimulation test was performed in 11 of 18 patients exhibiting low ACTH/cortisol levels, and ACTH deficiency was confirmed in all of them. A total of five patients underwent pituitary MRI before corticosteroid replacement therapy, and only one patient exhibited pituitary enlargement. No patient experienced primary adrenal deficiency. A total 16 of 18 patients with ACTH deficiency from hypopituitarism/hypophysitis experienced symptoms, including anorexia in 11, fatigue in 8, and headache, nausea, fever, and syncope in 2 patients (Figure 1a). Two patients exhibited TSH deficiency, and ACTH deficiency co-occurred in both. No patient had symptomatic hypogonadism, while only one patient exhibited decreases in LH and FSH levels. Hyperthyroidism or thyroiditis occurred in 17 patients (42%), 2 of whom were Grade 3 (Table 2 and Appendix A). Twelve of those seventeen patients were asymptomatic, while fever, edema, and fatigue were experienced by three, two, and one patient, respectively (Figure 1b). Primary hypothyroidism occurred in nine patients (22%), none of whom had symptoms. No patient developed diabetes mellitus.

### 3.4. Non-Endocrine irAEs

Non-endocrine irAEs were observed in 22 patients (54%). Skin-related irAEs occurred in 16 patients, including rash in 13, pruritus in 2, and hypopigmentation in 1 patient; only 1 patient had Grade 3 AEs (Table 2). Pneumonitis occurred in five patients, three of whom had Grade 3 AEs. Three patients experienced increases in alanine aminotransferase levels, two of whom had Grade 3 AEs. Three patients experienced increases in creatinine levels or acute kidney injury, one of whom had a Grade 4 AE. One patient each experienced Grade 3 myocarditis, Grade 2 colitis, Grade 2 radiculitis, and Grade 2 arthritis. No patients developed myositis.

### 3.5. Onset of irAEs

A total 76 irAEs were diagnosed from 41 patients following nivolumab plus ipilimumab therapy, in which 49 (64%), 19 (25%), and 8 (11%) occurred <3, 3–6, and ≥6 months after initiating nivolumab plus ipilimumab. In endocrine irAEs, hyperthyroidism, primary hypothyroidism and hypopituitarism occurred after a median of 1.4 (0.7–6.3 months), 2.1 (0.7–11.1 months), and 2.9 months (1.4–15.8 months), respectively (Figure 2). The median onset time of rash was 1.2 months (0.1–6.2 months). Pneumonitis occurred later, i.e., after a median of 4.7 months (1.1–6.4 months).

### 3.6. Co-Occurrence and Sequence of Endocrine irAEs

The co-occurrence of two or more endocrine irAEs was frequently observed (Figure 3). Co-occurrence of hyperthyroidism and hypopituitarism occurred in nine patients, eight of whom hyperthyroidism occurred first followed by hypopituitarism (Figure 4a,b). Co-occurrence of hyperthyroidism and primary hypothyroidism occurred in eight patients; hyperthyroidism occurred first, followed by hypothyroidism, in all of these cases (Figure 4c,d).

### 3.7. Association between Endocrine and Non-Endocrine irAEs

Endocrine irAEs sometimes co-occurred with non-endocrine irAEs other than skin-related irAEs (Figure 3). The incidence of irAEs other than skin-related irAEs increased with the number of endocrine irAEs per patient (7/30/41% of any grade, *p* = 0.10; 0/10/35% of ≥Grade 3, *p* = 0.027 in patients with zero/one/and two or more endocrine irAEs; Figure 5a,b). However, the number of endocrine irAEs was not statistically associated with skin-related irAEs (Figure 5c,d).

### 3.8. Assessing Predisposing Factors for High Grade irAEs

We assessed predisposing factors for ≥Grade 3 irAEs. However, age, sex, prior nephrectomy, IMDC risk, or histology of RCC did not differ significantly between patients with and without ≥Grade 3 irAEs (Table 3). Additionally, the number of doses of nivolumab plus ipilimumab did not differ significantly between the two groups (3.4 ± 1.0 in ≤Grade 2 or no irAEs versus 3.4 ± 0.93 in ≥Grade 3 irAEs, *p* = 0.84).

### 3.9. Corticosteroid Therapy and Thyroid Hormone Replacement Therapy

Corticosteroid replacement therapy was introduced in all 18 patients with secondary adrenal insufficiency due to ACTH deficiency/ hypopituitarism. Daily 15–300 mg (median 90 mg) hydrocortisone or 30–40 mg (median 40 mg) prednisolone was introduced as an initial therapy, depending on the severity of symptoms or co-occurrence of other irAEs. All of them continued corticosteroid replacement therapy, daily 15–20 mg hydrocortisone in 17 patients, and 40 mg prednisolone in one patient, as a maintenance therapy for a median of 8.3 months (1.0–18.3 months). Corticosteroid therapy to suppress inflammatory irAEs, such as pneumonitis or myocarditis, was introduced in six patients; only one patient discontinued corticosteroid therapy, while the other five continued low-dose corticosteroid therapy at the last follow-up, for a median of 8.7 months (4–17.2 months). Levothyroxine was introduced in all 9 patients with primary hypothyroidism; it was discontinued in 1 patient after recovery of thyroid hormone at 6.5 months, while the other 8 patients was still on levothyroxine at the last follow-up, for a median of 7.9 months (0.9–17.4 months).

## 4. Discussion

In this study, endocrinopathy, including hypopituitarism/ hypophysitis, hyperthyroidism, and primary hypothyroidism, was frequently observed (66%) in “real-world” Japanese RCC patients following nivolumab plus ipilimumab therapy. In the CheckMate 214 study, endocrine irAEs occurred in 33% of patients, and the incidence rates of hypothyroidism, hyperthyroidism, adrenal insufficiency, and hypophysitis were 16%, 11%, 5%, and 4%, respectively [13]. In our cohort, hypothyroidism, hyperthyroidism, and hypopituitarism with secondary adrenal insufficiency occurred in 22%, 41%, and 44% of patients, respectively, which was much higher than the incidence in the CheckMate 214 study. In total, 41% of patients discontinued nivolumab plus ipilimumab therapy permanently or temporarily due to irAEs in this study.

### 4.1. Hypophysitis and Hypopituitarism

Hypopituitarism due to hypophysitis was one of the most frequently observed irAEs following nivolumab plus ipilimumab therapy in this study. In the study of De Filette et al. [8], hypophysitis was more common in patients treated with ipilimumab (CTL4 inhibitor) (5.6%) compared to PD-1 inhibitors (0.5–1.1%), and combination therapy was associated with a high incidence (8.8–10.5%) of hypophysitis. Patients with hypophysitis present with symptoms related to mass effect from pituitary gland enlargement, such as headache and visual symptoms, due to compression of optic and/or cranial nerves, and pituitary dysfunction [14]. Immunotherapy-associated hypophysitis often presents with headache and anterior hypopituitarism, and the degree of pituitary enlargement is typically mild [15]. In this study, most patients with hypopituitarism/ hypophysitis presented with symptoms of adrenal insufficiency, such as anorexia and fatigue, while only 11% of those experienced headache, and no patients experienced visual symptoms (Figure 1a). Additionally, pituitary gland enlargement was observed in only one of five patients who underwent pituitary MRI, indicating that hypopituitarism/ hypophysitis following nivolumab plus ipilimumab might be less inflammatory and lack of mass effect from pituitary gland enlargement. All cases with hypopituitarism showed ACTH deficiency, while TSH deficiency and hypogonadism were rarely seen. Function of thyrotroph and gonadotroph axis may be restored, but not of corticotroph axis; thus, permanent replacement of corticosteroids is sometimes needed [16]. In line with this, all patients with ACTH deficiency required permanent corticosteroid replacement in this study. Faje et al. [15] reported that older age and male gender may be risk factors for the development of hypophysitis with anti-CTLA-4 medications. However, the patients with and without hypopituitarism/hypophysitis in this study did not differ significantly in age or sex (unpublished data). CTLA4 is expressed in cells in the anterior pituitary; its expression level varies greatly among individuals, indicating that the administration of CTLA4 inhibitors can cause complement activation in macrophages and infiltration of autoreactive T cells into the anterior pituitary in some patients [17,18]. In T cells, expression levels of CTLA-4 differ among single-nucleotide polymorphisms in the CTLA-4 gene, which might be associated with a better response to CTLA-4 blockade [19,20]. CTLA-4 genotyping might help stratify patients undergoing immunotherapy not only in terms of their response to therapy [21] but also the risk of developing irAEs, including hypophysitis.

### 4.2. Thyroid Dysfunction

Thyroid dysfunction was frequently observed following nivolumab plus ipilimumab therapy in this study. Thyroid dysfunction is the first or second most frequent irAE in patients treated with ICIs [8]. Thyroid disorders can occur in patients treated with ipilimumab (1–6%) but are more common in patients treated with anti-PD-1 (up to 40% of cases) [22]. Nivolumab plus ipilimumab increases the incidence of hypothyroidism and hyperthyroidism in melanoma [23]. In this study, immune-related hyperthyroidism/thyroiditis manifested as early onset thyrotoxicosis, which was largely asymptomatic (Figure 1b), and developed into hypothyroid or euthyroid thereafter (Figure 4c,d). Interestingly, when hyperthyroidism and hypopituitarism co-occurred in the same patient, hyperthyroidism usually occurred first (Figure 4b). The pathogenesis of thyroid disorders associated with immune check inhibitors is not completely known. However, in a case series of nivolumab-induced thyroiditis, PD-L1 and PD-L2 were found in normal thyroid glands, which suggests that administration of PD-1 inhibitors can disrupt the interaction between PD-1 on T cells and PDL-1/2 on thyrocytes, leading to T-cell activation against thyroid antigens [24,25].

### 4.3. Non-Endocrine irAEs

Skin-related irAEs were the most frequently observed non-endocrine irAEs and were often not severe in this study. Non-endocrine irAEs other than skin-related irAEs, including pneumonitis, hepatitis, myocarditis, and kidney injury, can be severe, sometimes requiring hospitalization and corticosteroid therapy. There was an association between endocrine irAEs and non-endocrine irAEs other than skin-related ones (Figure 5a,b). When our patients experienced two or more endocrine AEs, they had a 35% chance of experiencing ≥ Grade 3 non-endocrine irAEs (Figure 5b). There were fewer, or a similar number, of non-endocrine AEs in our cohort compared to the CheckMate 214 [13]. The median duration of nivolumab plus ipilimumab exposure (2.9 months) in our cohort was shorter than that in the CheckMate 214 study (7.9 months). Furthermore, corticosteroid replacement therapy was introduced in 18 of 41 patients for secondary adrenal insufficiency, which may help reduce non-endocrine irAEs.

### 4.4. Ethnic Differences in the Incidence and Severity of irAEs

The incidence and severity of irAEs may differ between Asian and Caucasian patients. Yang et al. [26] demonstrated that the incidence of selected AEs, including hypothyroidism, was significantly different between Asian and Western/international populations. In studies of non-small cell lung cancer conducted in Japan and Korea, nivolumab was associated with a higher incidence of treatment-related AEs compared to the CheckMate 057 and 017 studies which predominantly recruited Caucasian patients [27,28,29,30]. The Asian studies also reported a higher rate of treatment-related AEs, which led to discontinuation of nivolumab (15–16%), compared to the CheckMate 057 and 017 studies (1–3%). A study assessing the population pharmacokinetics of nivolumab demonstrated that nivolumab exposure was 46% higher in Asian patients due to lower body weight compared to non-Asian [31]. However, in the CheckMate 214 study, the incidence of irAEs in the Japanese subgroup was comparable to the entire cohort [13,32]. It is too early to definitively conclude that the incidence and severity of irAEs differ between Asian and Caucasian patients.

### 4.5. Difference in the Number of Endocrine irAEs between CheckMate 214 and This Study

The difference in the number of endocrine irAEs between CheckMate 214 and this study may stem from the difference in laboratory assessment strategy, i.e., routine laboratory assessments, including TSH, fT4, and fT3, were performed up to Cycle 3, during nivolumab-ipilimumab therapy, and additional measurement of these laboratory values were performed only as clinically indicated or to comply with local regulations in CheckMate 214 [2], while routine laboratory assessments were performed every or every other cycles in this study. Additionally, screening for central endocrine toxicity is not always required in clinical trials; therefore, some oligosymptomatic or transitory cases may have been missed. In CheckMate 214, 37% and 20% of patients experienced fatigues and nauseas in patients treated with nivolumab plus ipilimumab [2], which are also frequently observed symptoms in endocrine irAEs (Figure 1); therefore, some of them might have endocrine irAEs. Furthermore, secondary adrenal insufficiency can be masked by corticosteroid therapy for other irAEs. We experienced a case (case JMU1 in Figure 3) who exhibited secondary adrenal insufficiency after discontinuation of prednisolone for pneumonitis (unpublished data). The patient experienced no symptoms of adrenal insufficiency while receiving prednisolone. In CheckMate 214, 35% of patients received high-dose glucocorticoids following nivolumab plus ipilimumab therapy, which might mask the presence of secondary adrenal insufficiency. The difference in age (68 years in this study versus 62 years in CheckMate 214) and ethnicity between two studies may be associated with the difference in incidence of endocrine irAEs.

The strengths of our study include the close clinical and biochemical monitoring of patients from multiple institutes. Moreover, endocrine irAEs were reviewed by an experienced endocrinologist. The limited sample sizes and follow-ups, as well as regional differences, were the main limitations of this study. In addition, the follow-up strategies somewhat differed among institutions. Furthermore, we did not identify predisposing factors for high grade irAEs. Further large-scale validation studies are necessary to confirm our results. However, the present study provided a real-world evidences on the safety profiles of the combination therapy. An accumulation of knowledge of the safety profiles enables the development of more appropriate strategies to prevent or manage ICI toxicities without compromising the success of cancer immunotherapy.

## 5. Conclusions

Endocrine irAEs, including hypopituitarism, hyperthyroidism, and primary hypothyroidism, may occur in a higher frequency in “real-world” RCC patients treated with nivolumab plus ipilimumab therapy compared to the CheckMate 214 study. Corticosteroid or thyroid hormone replacement therapy is usually required on a permanent basis in patients with ACTH deficiency or primary hypothyroidism. Careful observation would be necessary in patients experiencing endocrine irAEs because endocrine irAEs frequently co-occur with other types of endocrine irAEs and/or high grade non-endocrine irAEs.

## Figures and Tables

**Figure 1 jcm-10-04767-f001:**
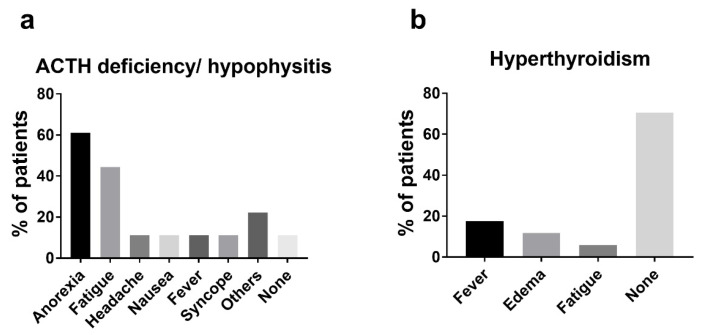
Symptoms in endocrine irAEs. (**a**) ACTH deficiency/ hypophysitis (*n* = 18); (**b**) hyperthyroidism (*n* = 17).

**Figure 2 jcm-10-04767-f002:**
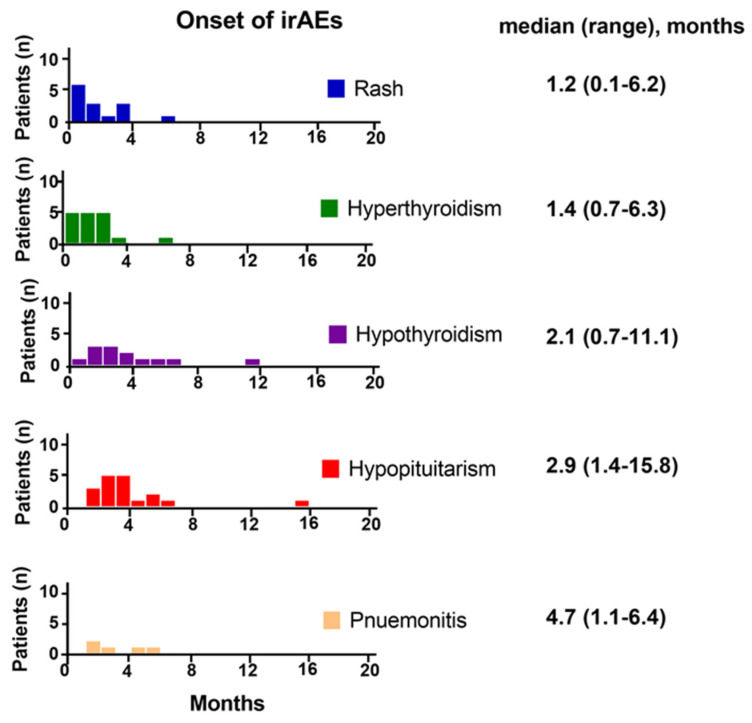
Onset times of each irAE. Blue, green, purple, red, and orange bars indicate the number of incidences of rash, hyperthyroidism, primary hypothyroidism, hypopituitarism, and pneumonitis, respectively, in each month following nivolumab plus ipilimumab initiation.

**Figure 3 jcm-10-04767-f003:**
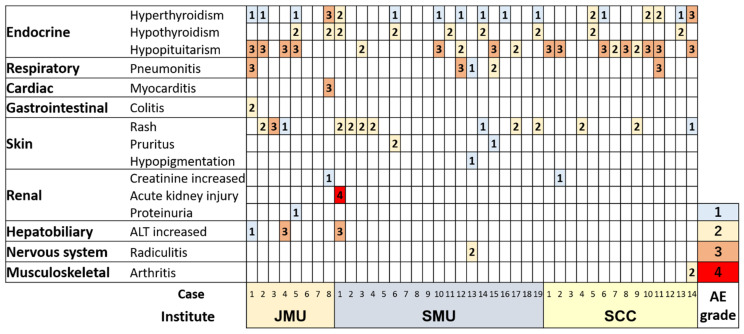
The grades of individual irAEs in each patient. JMU: Jichi Medical University Saitama Medical Center; SMC: Saitama Medical Center Saitama Medical University, SCC: Saitama Cancer Center; AE: adverse events.

**Figure 4 jcm-10-04767-f004:**
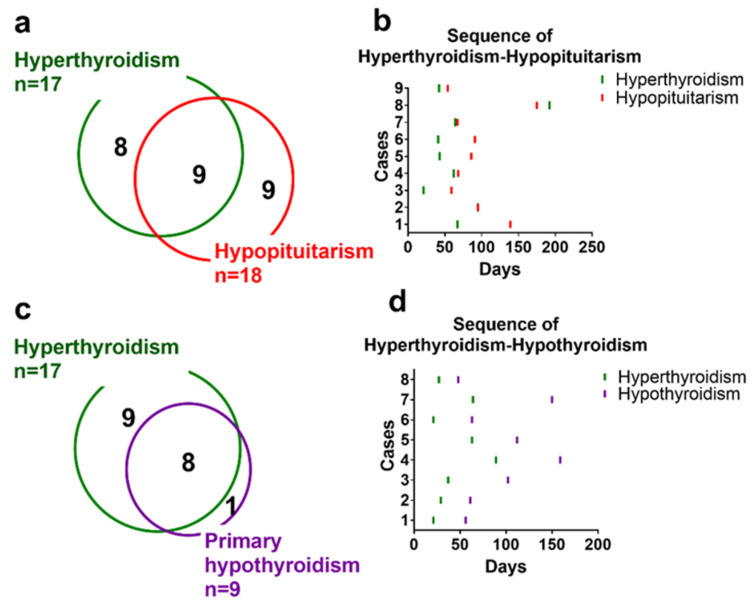
Co-occurrence of endocrine irAEs. Co-occurrences and sequences of hyperthyroidism with hypopituitarism (**a**,**b**) and primary hypothyroidism (**c**,**d**).

**Figure 5 jcm-10-04767-f005:**
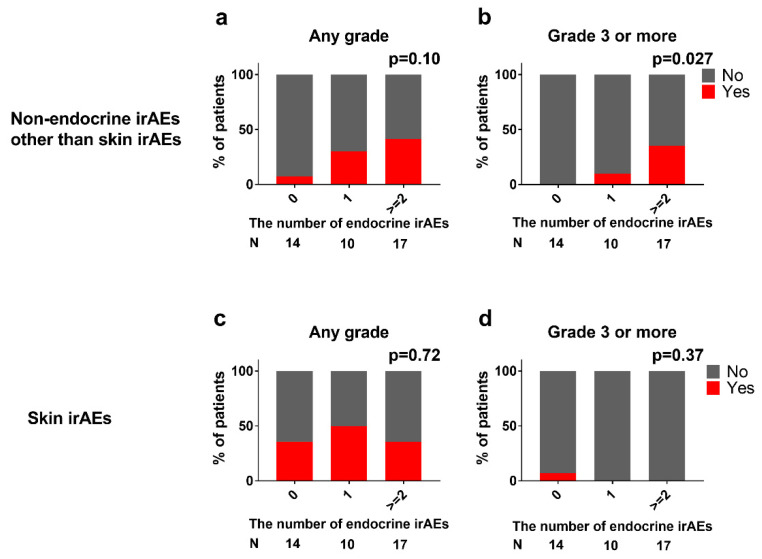
Association of endocrine irAEs with non-endocrine irAEs. The number of endocrine irAEs was associated with non-endocrine irAEs other than skin-related irAEs of any grade (*p* = 0.10 (**a**)) and Grade 3 or more (*p* = 0.027 (**b**)) but not with skin-related irAEs of any grade (*p* = 0.72 (**c**)) nor Grade 3 or more (*p* = 0.37 (**d**)).

**Table 1 jcm-10-04767-t001:** Patients’ characteristics.

		N	(%)
Age	Median (range)	68	(44–87)
Sex	Man	28	(68)
	Woman	13	(32)
Prior Nephrectomy	Yes	17	(41)
	No	24	(59)
IMDC risk	Intermediate	15	(37)
	Poor	26	(63)
Histology	Clear cell	32	(78)
	Non-clear cell	7	(17)
	Unknown	2	(5)

IMDC: International Metastatic RCC Database Consortium.

**Table 2 jcm-10-04767-t002:** The number and grades of irAEs following nivolumab plus ipilimumab therapy.

			Grade 1–2		Grade 3–4		All Grade	
			N	(%)	N	(%)	N	(%)
Endocrine	Pituitary	Hypopituitarism	4	(10)	14	(34)	18	(44)
	Thyroid	Hyperthyroidism/ thyroiditis	15	(37)	2	(5)	17	(41)
		Primary hypothyroidism	9	(22)	0	(0)	9	(22)
Non-endocrine	Respiratory	Pneumonitis	2	(5)	3	(7)	5	(12)
	Cardiac	Myocarditis	0	(0)	1	(2)	1	(2)
	Gastrointestinal	Colitis	1	(2)	0	(0)	1	(2)
	Renal	Acute kidney injury/ creatinine increased	2	(5)	1	(2)	3	(7)
		Proteinuria	1	(2)	0	(0)	1	(2)
	Hepatobiliary	Hepatitis/ ALT increased	1	(2)	2	(5)	3	(7)
	Nervous system	Radiculitis	1	(2)	0	(0)	1	(2)
	Skin	Rash/dermatitis	12	(29)	1	(2)	13	(32)
		Pruritus	2	(5)	0	(0)	2	(5)
		Hypopigmentation	1	(2)	0	(0)	1	(2)
	Musculoskeletal	Arthritis	1	(2)	0	(0)	1	(2)

ACTH: adrenocorticotropin hormone, TSH: thyroid stimulating hormone, ALT: alanine aminotransferase.

**Table 3 jcm-10-04767-t003:** Assessment of predisposing factors for high grade irAEs.

		≥Grade 3 irAEs (N = 24)	≤Grade 2 or No irAEs (N = 17)	*p* Values
Age	median (range)	68 (44–87)	69 (56–83)	0.43
Sex	Man	15	13	0.50
	Woman	9	4	
Prior Nephrectomy	Yes	7	10	0.11
	No	17	7	
IMDC risk	Intermediate	14	12	0.51
	Poor	10	5	
Histology	Clear cell	18	15	0.40
	Non-clear cell	4	2	
	Unknown	2	0	

IMDC: International Metastatic RCC Database Consortium.

## Data Availability

The data are available on request by contacting the corresponding author (S.W.).

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
