# Peer review of "Real-World Incidence of Immune-Related Adverse Events Associated with Nivolumab Plus Ipilimumab in Patients with Advanced Renal Cell Carcinoma: A Retrospective Observational Study"

_jcm, 2021, doi:10.3390/jcm10204767_

Round 1

Reviewer 1 Report

The present manuscript has an interesting topic, about the  incidence of immune-related adverse events associated with nivolumab plus ipilimumab in patients with metastatic renal cell carcinoma. The title is clear and easy to be individualized on literature research. The abstract is well structured, but the aim of the study is not presented.

The Introduction is well documented, but the aim of the study is not stated in the last paragraph of the chapter. More data about side effects use of Nivolumab plus ipilimumab in other neoplastic diseases should be presented.

Material and Methods are well conceived, to make the study reproducible, but more data about study design, more clear inclusion and exclusion criteria should be presented. The objectives of the study should be included in Introduction Section. The statistically analysis is shortly presented. How was the association between endocrine and non-endocrine irAEs  statistically assesed?

The Results cover all the required fields.

Discussions are well conceived and support the Results. The manuscript includes some phrases about the study limitations but the authors do not mention the impact of the study on the literature research. 

The Conclusions reflect the idea of the title and the manuscript presents a recent bibliography, with a reasonable number of titles.

Author Response

We would like to thank you very much for reviewing our manuscript and providing us insightful comments. We revised our manuscript based on your and another reviewer’s comments. We also made some minor changes for parts we realized that should be revised. The parts being changed are yellow highlighted. We put point-by-point responses as below. We hope you could check these out.

(Abstract)

#1 The abstract is well structured, but the aim of the study is not presented.

Response: Thank you very much for your comment and we agree with this comment. We added one sentence about the aim of this study into the abstract (Line 17-18).

(Introduction)

#2 The Introduction is well documented, but the aim of the study is not stated in the last paragraph of the chapter.

#3 More data about side effects use of Nivolumab plus ipilimumab in other neoplastic diseases should be presented.

Response: Thank you very much for these comments. We revised the Introduction section accordingly and added the safety outcomes of the combination of nivolumab + ipilimumab in melanoma (Line60-63 and Line 66-68).

(Material and Methods)

#4 More data about study design, more clear inclusion and exclusion criteria should be presented.

Response: Thank you very much for this comment. We included all patients who received nivolumab plus ipilimumab for previously untreated metastatic and/or locally advanced unresectable RCC in three institutes. There was no exclude criteria and we assessed all of them. We revised our manuscript accordingly (Line 77 and Line 80).

#5 The objectives of the study should be included in Introduction Section.

Response: Thank you very much for this comment. We added the objectives of the study in the Introduction section. We do think that the objectives of the study need to be also specified in the Materials and Methods section and therefore we did not make any changes in this point.

#6 The statistically analysis is shortly presented. How was the association between endocrine and non-endocrine irAEs statistically assessed?

Response: Thank you very much for this comment. We specified how statistical analyses were performed (Line126-128).

(Discussions)

#7 The manuscript includes some phrases about the study limitations but the authors do not mention the impact of the study on the literature research.

Response: Thank you very much for this comment. We totally agree with this comment. We added the strength of the present study and how this study impacts on the literature researches and/or clinical management (Line 367-376).

Reviewer 2 Report

    In this manuscript, authors discovered the incidences of immune-related adverse events in RCC patients treated with nivolumab plus ipilimumab. The findings are potentially significant that might attract attention from both physicians and basic researchers. However, several concerns are also raised after evaluating the content of this manuscript.

Major comments

  1. It is obscure to enroll those RCC patients specific with cancer metastasis.
  2. Other potential confounding factors other than nivolumab and ipilimumab for causing irAEs were not examined by multivariate statistics including gender, age, RCC histology and TNM classification.
  3. The association between the number of dose treatments and endocrine irAEs is not clear.

Minor comments

  1. References should be provided and cited (line#43, Nivolumab plus ipilimumab is approved as a first-line treatment for RCC).
  2. An extra space after line#211 should be removed.

Author Response

We would like to thank you very much for reviewing our manuscript and providing us insightful comments. We revised our manuscript based on your and another reviewer’s comments. We also made some minor changes for parts we realized that should be revised. Additionally references were added accordingly. The parts being changed are yellow highlighted. We put point-by-point responses as below. We hope you could check these out.

(Major comments)

#1 It is obscure to enroll those RCC patients specific with cancer metastasis.

Response: Thank you for this comment. Actually most of patients exhibited metastatic diseases at the initiation of this therapy while some of them exhibited lymph node involvements or a locally advanced renal tumor. We revised our manuscript accordingly (Line 20, 77, 138-141). We also changed the title of this study accordingly.

#2 Other potential confounding factors other than nivolumab and ipilimumab for causing irAEs were not examined by multivariate statistics including gender, age, RCC histology and TNM classification.

#3The association between the number of dose treatments and endocrine irAEs is not clear.

Response: Thank you very much for these comments. We added one section assessing predisposing factors for high grade irAEs. We performed univariate analysis, demonstrating that there was no predisposing factors for high grade irAEs, and therefore we did not perform multivariate analysis (Line 233-240 and Table 3, and Line 372-373).

(Minor comments)

  1. References should be provided and cited (line#43, Nivolumab plus ipilimumab is approved as a first-line treatment for RCC).
  2. An extra space after line#211 should be removed.

Response: Thank you very much for these comments. We revised our manuscript accordingly (Line 47-48).

Round 2

Reviewer 1 Report

The manuscript should be accepted for publication.

Author Response

I would like to thank you very much.

Reviewer 2 Report

The manuscript has been improved to warrant publication in JCM.

Author Response

I would like to thank you very much.

We corrected grammatical and typo errors as much as possible.